# Teaming up Radio and Sub-mm/FIR Observations to Probe Dusty Star-Forming Galaxies

Meriem Behiri [1,2,*], Marika Giulietti [1,2], Vincenzo Galluzzi [2,3], Andrea Lapi [1,2,4,5], Elisabetta Liuzzo [2] and Marcella Massardi [1,2]

1   Scuola Internazionale Superiore di Studi Avanzati, Via Bonomea 265, 34136 Trieste, Italy; mgiuliet@sissa.it (M.G.); lapi@sissa.it (A.L.); massardi@ira.inaf.it (M.M.)
2   INAF-Istituto di Radioastronomia-Italian ALMA Regional Centre, Via Gobetti 101, 40129 Bologna, Italy; vincenzo.galluzzi@inaf.it (V.G.); liuzzo@ira.inaf.it (E.L.)
3   INAF-Osservatorio Astronomico di Trieste-Italian Centre for Astronomical Archives, Via Giambattista Tiepolo 11, 34131 Trieste, Italy
4   IFPU—Institute for Fundamental Physics of the Universe, Via Beirut 2, 34014 Trieste, Italy
5   INFN-Sezione di Trieste, Via Valerio 2, 34127 Trieste, Italy
*   Correspondence: mbehiri@sissa.it

**Abstract:** In this paper, we investigate the benefits of teaming up data from the radio to the far-infrared (FIR) regime for the characterization of dusty star-forming galaxies (DSFGs). These galaxies are thought to be the star-forming progenitors of local massive quiescent galaxies and to play a pivotal role in the reconstruction of the cosmic star formation rate density up to high redshift. Due to their dust-enshrouded nature, DSFGs are often invisible in the near-infrared/optical/UV bands. Therefore, they necessitate observations at longer wavelengths, primarily the FIR band, where dust emission occurs, and the radio band, which is not affected by dust absorption. Combining data from these two spectral windows makes it possible to characterize even the dustiest objects, enabling the retrieval of information about their age, dust temperature, and star-formation status, and facilitates the differentiation between various galaxy populations that evolve throughout cosmic history. Despite the detection of faint radio sources being a challenging task, this study demonstrates that an effective strategy to build statistically relevant samples of DSFGs would be reaching deep sensitivities in the radio band, even restricted to smaller areas, and then combining these radio observations with FIR/submm data. Additionally, this paper quantifies the improvement in the spectral energy distribution (SED) reconstruction of DSFGs by incorporating ALMA band measurements, in particular, in its upgraded status thanks to the anticipated Wideband Sensitivity Upgrade.

**Keywords:** extragalactic radio sources; sub-mm galaxies; star formation; galaxy evolution





## 1. Introduction

Dust and synchrotron emissions draw the early evolutionary stages of galaxies: they depict the relative importance of star formation and nuclear activity and the imprint of stellar evolution in shaping the galactic components. The dusty star-forming galaxies (DSFGs) label classifies objects typically selected in the sub-millimeter/far-infrared (sub-mm/FIR) domain (and therefore sometimes tagged as "Submm galaxies" or SMGs; e.g., [1–3]) with a rich abundance in dust $M_{dust} \gtrsim 5$–$20 \times 10^8 \, M_\odot$ and relatively high star formation rates (SFRs) $\gtrsim 10^2 \, M_\odot/yr$ (see [4] for a comprehensive review).

DSFGs are found to be abundant at high redshift $z \gtrsim 1$ (e.g., [5]) and have been detected up to $z \sim 6$ [4,6–8]. They provide a substantial contribution to the cosmic star formation history at $z \sim 1$–4. Because of their huge dust content, these objects are heavily obscured in optical bands and extremely bright in sub-mm/FIR bands where the light of newborn hot stars, reprocessed by dust grains around star formation regions, is re-emitted. The presence of dust and, thus, of high obscuration and sub-mm/FIR luminosity is a

strong indicator of ongoing active star formation, often accompanied by another process that is fueled by large gas reservoirs: a fast growth of a central supermassive black hole (SMBH) buried in the dust component. Eventually, its emitted power through winds and/or jets may cause negative feedback that could sweep away part of the interstellar medium. The star formation may thus be quenched and dust removed after a relatively short timescale $\lesssim 1$ Gyr, when the active nucleus may shine as an optical unobscured quasar. Subsequently, the galaxy is left devoid of gas, and the stellar population evolves passively; thus, DSFGs are thought to be the progenitors of local massive quiescent galaxies [9].

This qualitative picture needs to be substantiated by high-quality multi-wavelength data, providing detailed information on the structure of DSFGs in different evolutionary phases. In particular, observations in the radio bands can play a key role in improving our knowledge of DSFGs. This regime has many advantages such as the following:

- it is not affected by dust obscuration; therefore, it allows study of even the most obscured galaxies;
- it tracks both star formation (supernova explosions and HII regions) and AGN activity;
- it allows the construction of wide-field surveys, thanks to wider primary beams (with respect to those of sub-mm observatories) and the possibility of combining them in mosaic mode over large areas;
- it is possible to reach high resolution (down to sub-arcsec, e.g., [10]) thanks to interferometry.

Radio observations associate DSFGs with the faint (sub-mJy) population of objects [10–13]. The most significant bias possibly affecting radio selection is the possible presence of AGN, but this problem can be avoided thanks to multi-wavelength ancillary data. By matching the available information in the radio and sub-mm/FIR regimes, an FIR-radio correlation (FIRRC) has been well established in the local universe [5,14–18], and it can be used to distinguish between radio luminous AGNs and star-forming galaxies. Moreover, radio emission is insensitive to dust absorption and hence constitutes a good indicator of the intrinsic SFR in galaxies [19,20]. When complemented with the FIRRC to exclude a substantial AGN contribution, radio observations can become efficient tracers of the obscured star formation in DSFGs, hence a suitable probe to obtain a comprehensive view of the cosmic star formation history up to very high redshift [6,21,22]. In fact, at high redshift, the FIRRC has demonstrated a straightforward match between the radio and FIR behaviour of optically/radio-selected quasars and DSFGs [23,24].

Furthermore, the radio-to-FIR spectral regime is also known as the "cosmological window", as the combination of CMB foregrounds reduces to a minimum: this is due to the steepening of the synchrotron at frequencies above a few tens of GHz and the increase in thermal dust emissions at even higher frequencies. Extragalactic point sources are the dominant foreground to CMB emission on small angular scales, albeit they remain a poorly observationally constrained contribution that affects the total intensity and polarized CMB power spectrum at higher multipoles. In this respect, once the radio-loud AGNs are masked using positions from low radio-frequency surveys, the contribution of DSFGs dominates the foreground emission on the smaller angular scales in the whole radio-to-FIR domain, emerging among the most critical contaminants in the component separation efforts for present and future CMB facilities (e.g., Simons Observatory, LiteBIRD, and CMB-S4), as it cannot be easily predicted (each source has a different spectral energy distribution and in the AGN cases could be variable), and constituting an unresolved background below the detection threshold: it is therefore extremely important to statistically characterize the DSFG population behaviour down to low flux densities over the whole spectral regime. Reasons for such a missing thorough multi-frequency characterization (over statistically significant samples) of DSFGs can be traced back to technical issues and biases that typically have affected observations so far, such as poor spatial resolutions (and limiting confusion level), shallow sensitivities, and limited multi-frequency coverage.

In this paper, we focus on the relevance of combining DSFG data in the FIR regime with high resolution (few arcseconds) and deep (down to µJy) blind radio surveys to tackle

such observational limitations (*cf.* Section 2). In Section 3, we evaluate the role that the Atacama Large Millimeter/sub-millimeter Array (ALMA) telescope may have in improving the knowledge of the overall DSFG population, both in its current disposal and after the expected Wideband Sensitivity Upgrade (WSU, [25]). Finally, we present some take-home messages in Section 4.

## 2. Radio-to-FIR Observations of DSFGs

The radio spectra of DSFGs are a combination of a flat free-free component from HII regions containing massive ionizing stars and a steep synchrotron component resulting from relativistic electrons accelerated by supernova remnants. An additional synchrotron emission from a small-scale jet or winds/outflows associated with the nuclear activity may be present with various (and variable) spectral behaviours (see [26]). As the frequency increases, the radio emission is progressively overwhelmed by the rising, grey-body component due to dust emission associated with star formation. On average, the frequency of this transition falls in the range $\sim$30–100 GHz, but it depends on redshift, galaxy age, and the relative role of nuclear activity and star formation. In fact, although X-ray follow-up observations of FIR-selected DSFGs at high redshift have clearly pinpointed the presence of heavily obscured, accreting central supermassive black holes (see [27]), their capability for driving appreciable radio emission is still to be assessed, especially in connection with galaxy properties (e.g., age, specific star-formation rate, obscuration, etc.).

Hence, a comprehensive description of this galactic population necessitates the combination of both the radio and FIR information by selecting samples in one electromagnetic regime and performing follow-ups in the other or by merging statistically significant population properties obtained from large-area surveys. However, the large number of parameters and processes involved makes extrapolations between different frequency domains extremely tough and at the limit of instrumental sensitivities, so that follow-ups in other bands of selected samples typically have low success rates.

### 2.1. DSFG in Radio Observations

Despite DSFGs constituting the bulk of sub-mJy radio source populations [13], they remain substantially unexplored because of limits in sensitivity and resolution of the current facilities and of the positive *k*-correction that hampers the possibility of reaching extremely high redshifts in the radio bands.

The expected contribution to the 1.4 GHz source counts of the different populations of extragalactic sources has been estimated by [28] (see their Figure 8). In particular, at sub-mJy fluxes, the counts are dominated by star-forming galaxies (radio-loud AGNs are largely subdominant), of which DSFGs constitute a relevant fraction; this is even more true focusing on progressively high-redshift sources. For example, a star-forming galaxy at redshift $z \approx 1$ ($\approx$3) with 1.4 GHz radio flux of $\approx$100 µJy would feature a 1.4 GHz radio luminosity of $\approx 10^{30}$ ($\approx 10^{31}$) erg s$^{-1}$ Hz$^{-1}$; assuming a standard FIR-radio correlation, this would correspond to a FIR luminosity of $\approx 10^{45}$ ($\approx 10^{46}$) erg s$^{-1}$, which in turn would represent an appreciable SFR of $\approx$30 ($\approx$300) $M_\odot$ yr$^{-1}$. Such high SFRs imply a rapid metal and dust enrichment that make these objects heavily obscured and, thus, proper DSFGs. Their bright FIR/submm emission can be exploited, via cross-matching with radio surveys, to distinguish them from low-redshift $z \lesssim 1$, less star-forming, and less obscured systems.

For reference, based on the source counts by [28], we estimated the observing time requested by the Australia Telescope Compact Array (ATCA[1]) to detect star-forming galaxies at 2.1 GHz, reporting it as a function of the flux density limit and survey area in Figure 1: the estimates consider the mosaicing conditions (Nyquist sampling) for areas larger than 1 field-of-view (FOV) but does not consider overheads, which could be as high as 40% because of RFI typically affecting this frequency band. For the same conditions, we estimated the number of expected sources in a given area as a function of the flux density limit (see Figure 2). It is clear that the shape of the differential source counts implies that to achieve a statistically significant number of SFGs, it is more efficient to reach deep

sensitivities even if over small areas. Nevertheless, high redshift sources remain rare, and only deep large area surveys can collect significant samples.

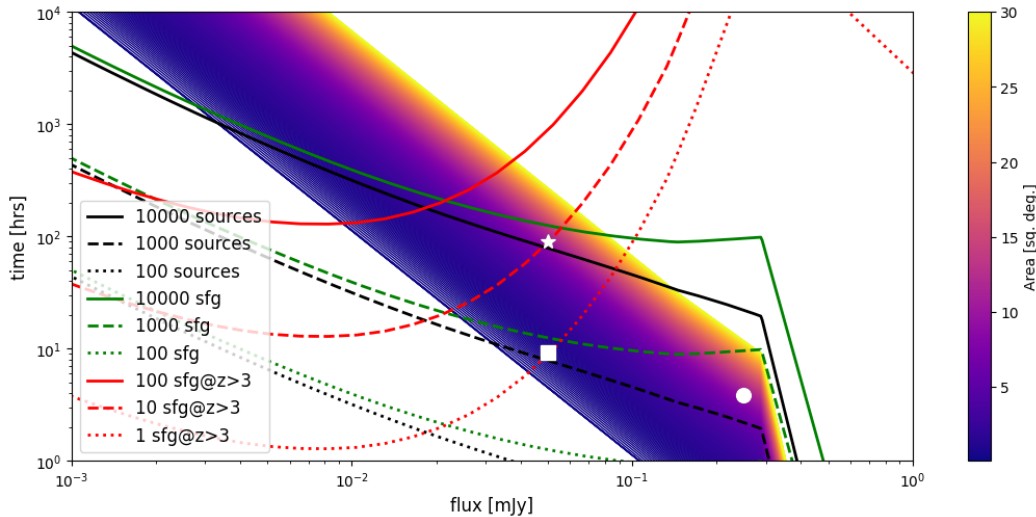

**Figure 1.** ATCA observing time estimated at 2.1 GHz requested to reach the values of flux density (displayed by the x-axis) over a given sky area (coloured scale). The estimate does not consider overhead time. For comparison, we also reported the time needed to perform a survey of 10,000 (black solid line), 1000 (black dashed line), or 100 (black dotted line) sources to observe 10,000, 1000, and 100 SFGs (green lines) or 100, 10, or 1 SFGs at redshift $z > 3$ (red lines). Three examples of surveys have been identified: (white square) a survey of 1 sqdeg down to a flux density limit of 0.0 mJy (at $5\sigma$ confidence level), (white star) a survey of 10 sqdeg down to a flux density limit of 0.05 mJy ($5\sigma$), and (white circle) a survey of 10 sqdeg down to a flux density limit of 0.25 mJy ($5\sigma$). No bias corrections have been introduced in the estimate of the source numbers.

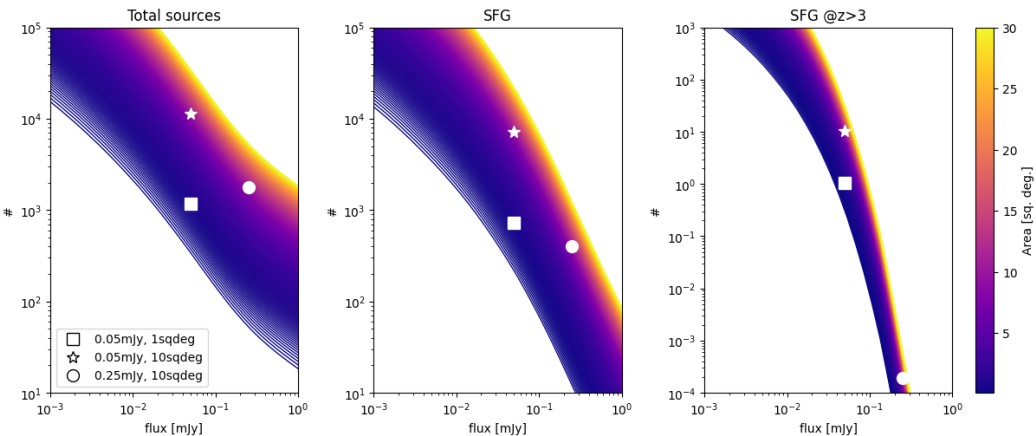

**Figure 2.** Number of sources expected with the ATCA at 2.1 GHz at the flux density (on the x-axis) in a given area (coloured scales) for the case of the total number of sources, the SFG only, and the DSFG at $z > 3$. The example cases are the same as in Figure 1. No bias corrections have been introduced in the estimate of the source numbers.

At frequencies $\nu \lesssim 5$ GHz ($\gtrsim 10$ cm in wavelength), the DSFG signal is dominated by the synchrotron emission associated with star formation [19,20]. The minimum, generated by the combination of fading synchrotron and rising dust signals, is expected to be located in the 30–100 GHz frequency ($\sim 3$–7 mm wavelength) range. Therefore, radio observations targeted to DSFGs are preferentially performed across $\sim 1$–3 GHz frequency range.

On the one hand, the FOV size scales as $\theta_{FOV} \sim \lambda / A$, where $\lambda$ is the observational wavelength and $A$ is the antenna(s) diameter size. Therefore, in the cm-wavelength domain, the larger FOV allows for a faster coverage of large areas of the sky, with respect to what

can be achieved at mm-wavelength bands. On the other hand, the resolution scales as $\theta_R \sim \lambda / B$, where $\lambda$ is the observational wavelength and $B$ is the maximum distance (i.e., the maximum baseline length), by taking into account all the couples of antennas in the array. Therefore, for a given telescope, the resolution gets worse in the cm-domain with respect to the mm-domain, with the consequence of possible effects of confusion and blending going to the deepest flux densities (i.e., Malmquist and Eddington biases).

Furthermore, the lowest frequencies are the most seriously affected by radio frequency interference (RFI) and human activities, raising dramatically the confusion level and the level of flagging that is needed to reach reliable detections when the observation requires high sensitivity, as for radio-faint DSFGs.

Other effects that must be accounted for are those introduced as a consequence of the imaging process for radio interferometers. Images are the result of an inverse Fourier transformation of the observed data (visibilities, edited to remove RFI and misbehaviours, if needed). The noise is non-Gaussian, with a pattern that is determined by the antenna configuration and the observing time allocation. The non-Gaussianity and the presence of noise features increase as the array is sparser or the observing time shorter or limited to small chunks. This issue implies that the reliability of a source might change depending on its position across the FOV and with respect to other brighter sources that may be present in the vicinity (e.g., within a few degrees). Techniques like self-calibration or source subtractions/peeling in the visibility domain can significantly improve the dynamic range in the image domain but are not always possible (e.g., not enough signal-to-noise ratio for self-calibration solutions to converge or poor sky/bright sources modelling). Continuum observation imaging combines visibilities over the whole bandwidth (i.e., multi-frequency synthesis) to achieve deeper sensitivities, but the telescope response may be significantly different across a wide bandwidth, as it scales linearly with the wavelength (as mentioned above for the overall FOV size). This fact can be significant at low frequencies, producing a chromatic aberration called "bandwidth smearing", which blurs images radially, altering source positions and flux density measurements.

For all these reasons, deep radio surveys have been typically limited to small areas, and uncertainty due to the confusing effects significantly hampers a comprehensive description of the sub-mJy population at cm-wavelengths. Most of the issues in modern surveys have been addressed by data processing software refinements, which for larger and larger facilities introduce computational complexity that make surveys critically resource demanding.

This situation is improving in the cm-domain thanks to the profusion of surveys already in progress or planned for the next years with SKA pathfinders and precursors (e.g., EMU, MIGHTEE, RACS, and VLASS), which will combine homogeneous low noise patterns, fast survey speed due to a large number of observing elements with high resolution over long baselines, and advanced processing techniques. Nevertheless, an issue concerning cross-matching with the FIR observations will remain, as explained in the next section.

In the mm-domain, the expected WSU for ALMA will dramatically increase the survey speed, allowing for relatively large areas of the sky to be covered in reasonable time, as we demonstrate later in this paper. The ALMA telescope frequency coverage from 30 to 950 GHz permits the combination of observations across the synchrotron- to dust-dominated regimes and the proper reconstruction of the spectral energy distribution (SED) of the observed DSFGs.

### 2.2. Cross-Matches of Radio and FIR Surveys

The Herschel surveys have depicted a general statistical description of the DSFG population in the FIR regime. Specifically, the Herschel Astrophysical Terahertz Large Area Survey (H-ATLAS) [29] is the largest, covering $600 \, \text{deg}^2$ in five photometric bands (100, 160, 250, 350, and 500 μm). A plethora of follow-ups of many small samples each selected with different criteria, for different purposes, have been observed in different settings with sub-mm telescopes (e.g., ALMA, PdB, and SPT). High-resolution and sensitivity follow-up

helped tremendously to achieve a robust characterization of the dust emission profiles and contributed to improving the redshift measurements for a currently consistent albeit inhomogeneous collection of DSFGs.

Furthermore, high-resolution follow-up pointed out that FIR surveys suffer from source blending due to the high density of sources occasionally enhanced by clustering. This is particularly relevant in the SPIRE 500 µm maps where the PSF is 35.2 arcsec$^2$: in several cases, the flux densities of multiple objects are mixed and associated to a single detection, confusing the determination of its dust properties and photometric redshift. This issue might limit the identification of counterparts in very densely populated surveys, like those reaching very deep levels at IR and optical wavelengths (e.g., Spitzer; see also [4,30]). ALMA high-resolution follow-ups at similar frequencies, VLA in the radio bands and Spitzer in the MIR, have helped in de-blending many sources (e.g., [31–33]). However, as the frequency gap between observations and follow-up increases, the cross-matches become more uncertain and biased.

On the one hand, radio follow-ups of FIR- or sub-mm-selected sources tend to be biased toward the silent radio objects, making it difficult to probe the overall faint population and quite often producing radio non-detections. On the other hand, the lack of sub-mm or FIR information for radio surveys results in poor classification and redshift determination of the detected sources via SED reconstruction (e.g., [13,34–36]). Therefore, an unbiased overview of the sub-mJy radio source population could be obtained by combining deep blind radio surveys with Herschel catalogues (e.g., [37]).

## 3. Radio-to-FIR SED Reconstruction

The availability of flux measurements in the radio and the FIR regime allows the reconstruction of the SED, hence the characterization of radio emissions, dust properties and an estimate of photometric redshift. The significance of such determinations depends on the accuracy of the available measurements and their distribution across the frequency range.

As mentioned above, while the radio regime observations are concentrated to frequencies $\lesssim 2$ GHz and FIR measurements associated with *Herschel* surveys, the millimetre-wavelength regime is mostly limited to the availability of ALMA follow-ups, which could complement the transition region between synchrotron and free-free in the radio and dust emission in the sub-mm domain.

To quantify the effect of the availability of ALMA measurements on the SED determination of sources, we have identified a region of the H-ATLAS South Galactic Pole (SGP) survey that has been observed with ATCA at 2.1 GHz down to 10 µJy sensitivity in the framework of the Serendipitous H-ATLAS Observation of Radio Extragalactic Sources (SHORES, Massardi et al. in prep) survey. Specifically, the sample is constituted of all the 60 sources detected at 2.1 GHz in ATCA observations that presented an H-ATLAS counterpart in about 2 deg$^2$ of the survey area.

We performed MCMC simulations of the plausible SED fitting by considering the combination of synchrotron (single steep spectrum power-law), free-free (single power-law with slope equal to 0.1), and dust (grey-body spectrum) emissions. Figure 3 demonstrates the improvement of adding ALMA points (specifically a detection in Band 1 [henceforth B1] at 39 GHz and/or a detection in Band 6 [B6] at 233 GHz) to the SED determination for one target representative of our sample. In each band, we considered the sensitivity levels attainable with ALMA observations in the current telescope status in less than 10 min corresponding to 20 µJy at 39 GHz and 40 µJy at 233 GHz.

For the same source, in Figure 4, we show that even in the case of a non-detection in ALMA data, the definition of the SED improves with respect to the situation without ALMA data.

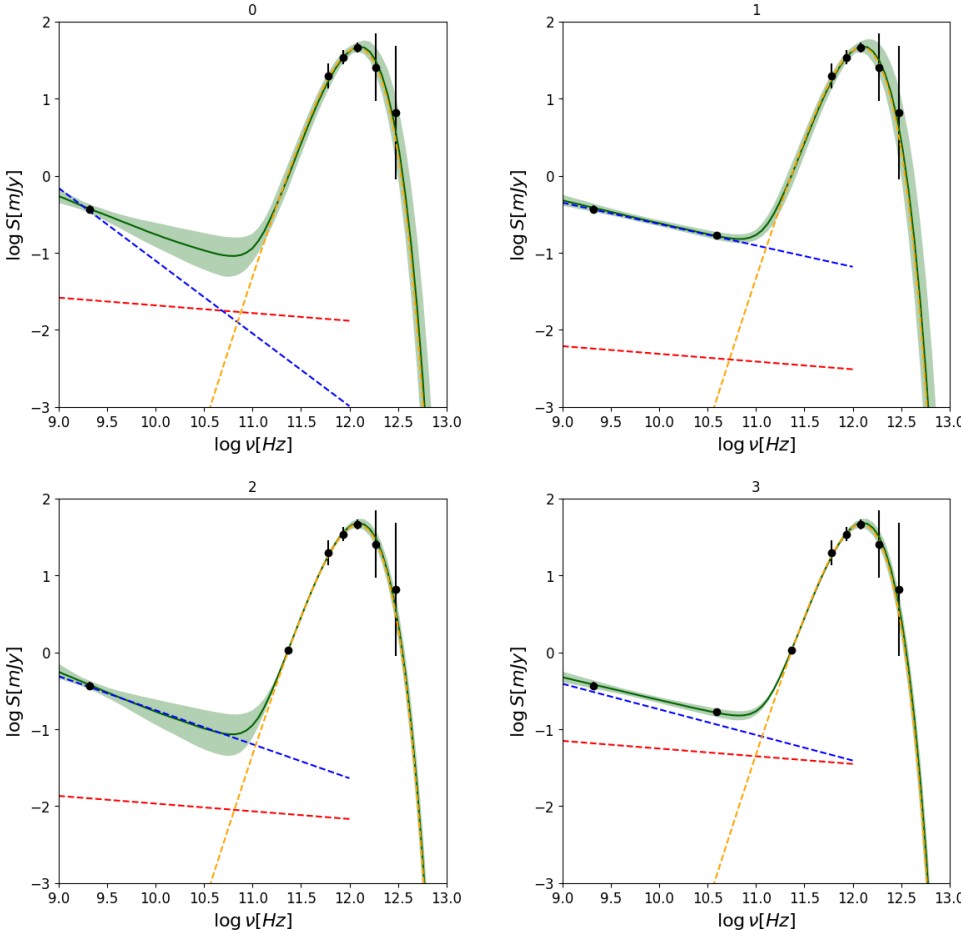

**Figure 3.** MCMC simulation of the best (solid green line) and 1-sigma confidence intervals (green shaded area) of possible fitting SED solutions for a representative source in our sample in the log frequency range 9–15 Hz corresponding to wavelengths between 5.5 and 1.5 μm. The top left panel shows the situation without any ALMA observations. The top right and bottom left panels show the improvement in the SED definition (i.e., a thinner green region) obtained by adding only B1 or B6 ALMA, respectively, considering noise of 20 μJy and 40 μJy in B1 and B6, respectively, attainable in less than 10 min on source with the current ALMA array. Finally, the bottom right panel shows the case of the combination of existing ATCA and H-ATLAS data with ALMA B1 and B6 measurements with less than 10 min on source at each band. To demonstrate how the best-fitting solution varies, dashed lines show the best-fitting synchrotron (blue), free-free (red), and dust emission (yellow).

Figure 5 presents the distributions of the relative errors for some of the parameters used for the fits. The medians of the distributions in all the cases are significantly lower if ALMA data are included in the SEDs.

The broadening of the bandwidths planned for the future ALMA WSU could shorten the observing time at least by a factor $\sqrt{2}$ and 2, respectively, in Band 1 and 6. Further factors could be gained by the digitalization schemes of the new correlator, reducing the surveying time by factors up to 9.6 [25]. This will strongly enhance the possibility of follow-up of statistically significant samples in reasonable times or covering in blind survey modes the regions already surveyed in the radio domain.

Therefore, the improvement in SED reconstruction that will be obtained with the ALMA observations suggested here will allow us to characterize the sub-mJy radio source population and to quantify the relative role of nuclear activity and star formation in galaxies at intermediate/high redshifts, improving over any past attempt thanks to the statistical significance of the samples and to its broad spectral coverage, including ancillary datasets.

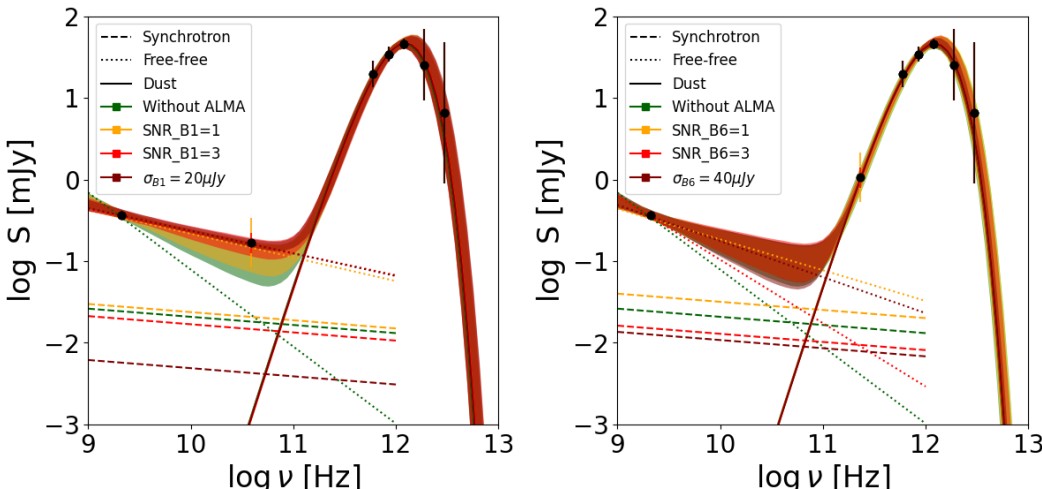

**Figure 4.** (**Left panel**) Comparison of the $1\sigma$ confidence levels from global SED fitting in case of no ALMA data (green shaded area) and three cases of B1 observations: a non-detection (upper limit at 150 μJy in the example, orange), a detection at $3\sigma$ significance ($150 \pm 50$ μJy, red), and a strong detection above $5\sigma$ significance (maroon) for the same source used as an example in Figure 3. As a reference, a 10 min observation in B1 with ALMA results in a noise level of $\sim$20 μJy. By using the same colour code, we also display how the best fitting for synchrotron (dotted lines), free-free (dashed lines), and dust (solid lines) emissions changes. We note that even non-detections provide precious indications, in particular for free-free and synchrotron components (as only the B1 is varying in this example), hence improving the global SED definition. (**Right panel**) The same as in the left panel, but varying the noise of the B6 detection: a 10 min observation in B6 with ALMA results in a noise level of $\sim$40 μJy.

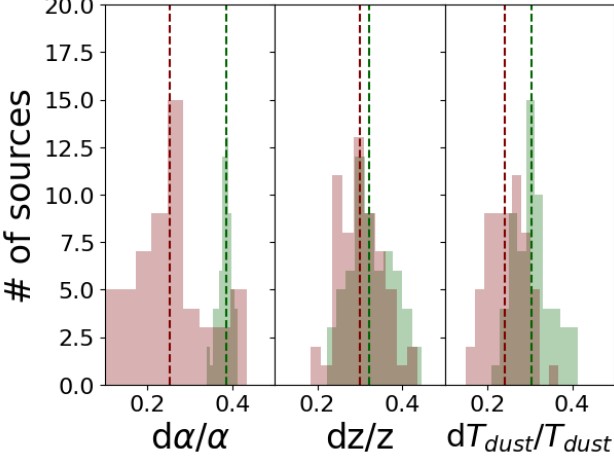

**Figure 5.** Distribution of the relative error in the parameter estimations for the synchrotron spectral index, redshift, and dust temperature without (green) and with (maroon) the estimated ALMA data for the 60 sources in our sample: the medians of the distributions (dashed lines) in all the cases are considerably lower if ALMA data are included in the SEDs. Clearly, in most cases, the synchrotron spectral index cannot be defined without an ALMA B1 point, giving an unrealistically narrow distribution, making any significance test unreliable.

## 4. Summary

In this paper, we described the relevance of teaming up DSFG data available from the radio-to-FIR regime to surpass the observational gaps in the characterization of source properties. In particular, we have stressed that, since radio and FIR emissions are strictly related via the mechanisms of star formation and AGN feedback operating in DSFGs, a comprehensive description of this galactic population can only be obtained by selecting

samples in the one regime and performing follow-up in the other or by merging statistically significant population properties obtained in large area surveys. However, DSFGs are typically faint in the radio domain, and observations in this regime have been so far mostly limited to small areas still susceptible to confusing effects. We have demonstrated that, due to the source count profiles, it is more efficient to attain deep sensitivities in the radio domain, even in limited areas, to achieve a statistically significant number of DSFGs, eventually complementing the source analysis by cross-matching the radio observations with ancillary datasets in the FIR domain, such as Herschel surveys.

Moreover, we have quantified the improvement in the SED reconstruction of sources observed in radio and Herschel bands by adding measurements in the ALMA bands, both in the current telescope conditions and after its expected major upcoming Wideband Sensitivity Upgrade. In the latter case, observing time will be reduced by factors up to 20; therefore, follow-ups of statistically significant samples will certainly improve our capabilities to comprehensively describe the DSFG population.

**Author Contributions:** Conceptualization: M.B., M.G., V.G., A.L., E.L. and M.M.; methodology: M.B., M.G., V.G. and M.M.; validation: E.L. and M.M.; and writing: M.M. and A.L. All authors have read and agreed to the published version of the manuscript.

**Funding:** This work was partially funded from the following projects: "Data Science methods for MultiMessenger Astrophysics & Multi-Survey Cosmology" funded by the Italian Ministry of University and Research, Programmazione triennale 2021/2023 (DM n.2503 dd. 9 December 2019), Programma Congiunto Scuole; Italian Research Center on High Performance Computing Big Data and Quantum Computing (ICSC), project funded by European Union—NextGenerationEU- and National Recovery and Resilience Plan (NRRP)—Mission 4 Component 2 within the activities of Spoke 3 (Astrophysics and Cosmos Observations); INAF Large Grant 2022 funding scheme with the project "MeerKAT and LOFAR Team up: a Unique Radio Window on Galaxy/AGN co-Evolution"; and INAF GO-GTO Normal 2023 funding scheme with the project "Serendipitous H-ATLAS-fields Observations of Radio Extragalactic Sources (SHORES)".

**Data Availability Statement:** Data are contained within the article

**Acknowledgments:** We acknowledge the anonymous referees for constructive comments and suggestions. We thank I. Prandoni for useful discussions.

**Conflicts of Interest:** The authors declare no conflicts of interest.

## Notes

[1] ATCA is a $6 \times 22$ m antennas array located in New South Wales (Australia) operating in continuum in 5 bands between 1.1 and 105 GHz with $2 \times 2$ GHz bandwidth, https://www.narrabri.atnf.csiro.au/observing/. URL accessed on 21 March 2024.

[2] This should be compared to the FWHM of PACS observations 11.4 and 13.7 arcsec at 100 and 160 μm and those of the other SPIRE bands, namely 17.8 and 24.0 arcsec, respectively at 250 and 350 μm.

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
