# Peer review of "Teaming up Radio and Sub-mm/FIR Observations to Probe Dusty Star-Forming Galaxies"

_galaxies, doi:10.3390/galaxies12020014_

Round 1

Reviewer 1 Report

Comments and Suggestions for Authors

This paper presents a case for studying Dusty Star-Forming Galaxies (DSFGs) based on the far-infrared to radio wavelength regime, which can allow to estimate important characteristics such as age, dust temperature and SF status. A particular attention is given to ALMA band measurements, considering its upcoming Wideband Sensitivity Upgrade. The paper further argues for the best strategy to build samples of DSFGs concerning area vs depth.

The results presented are informative and useful for the community. At points, however, the paper appears to have been quickly written and lacking the quality required for publication. I believe these are easily fixable faults, and I try to list them below. I do have one main concern, which is the use of Mancuso+17 (which addresses the general star-forming population) for the DSFGs of this paper. I detail this concern below.

===== 

1. (l. 29) "[DSFGs]... are the main contributors to the total star formation history (SFH) at 1 ≲ z ≲ 4 and have been detected up to z ∼ 6." - this is a strong statement that should have a reference. 

2. (l. 33) "The presence of dust, and, thus, of high obscuration and submm/FIR luminosity, is, in fact, a signature of ongoing active star formation." - although this may be correct, the sentence ignores the possible presence of an AGN, which is found in many SMGs, and to which the authors refer immediately afterwards. I believe the sentence could be more accurately written in a more open way, such as "... is a strong indicator of ongoing active star formation, often accompanied by..."

3. (l. 37) "its emitted power through winds and/or jets will cause strong negative feedback" --- probably better as "may cause negative impact", as many cases are known of positive feedback.

4. (l. 38) "The star formation is thus quenched" --- accordingly, probably better as ""The 38 star formation may thus be quenched"

5. (l. 50) "wider beam size and the possibility of using the mosaic technique;" -- I would not say this, as (a) the "wider beam size" is, more correctly, a "wider primary beam, or field-of-view, size" and (b) the mosaic technique is not exclusive to observations in radio bands...  

6. (l. 52) "it is possible to reach high resolution (up to sub-arcsec, e.g. [6])" -- probably "(down to sub-arcsec..."?

7. (l. 54) "Radio observations ([7], [6], [8]) associated DSFGs with the faint (sub-mJy) population of objects. " -- there's a missing reference here, as I believe neither [6], nor [7] nor [8] establish any link of sub-mJy radio sources to (mm/FIR selected) DSFGs. 

8. (l. 58) "FIR-radio correlation (FIRRC) has been well established in the local universe ([9];[10]; [11]; [12]; [13]; [5]), " -- I don't think reference [9] has any content about the FIRRC, and neither [5]...?

9. (l. 59) "between radio luminous AGNs and the star-forming galaxies." -- "between radio luminous AGNs and star-forming galaxies."?

10. (l. 60) "Thus, this relation has addressed the radio emission as an efficient tracer of obscured star formation in dusty galaxies ([14])," -- two notes here. I don't think "addressed" is the correct word here; I also don't think reference [14] is an adequate reference here. Probably you should use a reference like Condon, 1992, for using radio as a SFR (dust-free) indicator in galaxies, and then eventually provide a more recent specific reference concerning the dust-obscured population?

11. (l.69) "Despite dominant foregrounds come from our Galaxy" -- check english please.

12. (l. 72) "In this respect, masked using positions from low radio frequency surveys" -- this sentence is not clear, probably due to the english construction. Please check.

13. (l. 98) "spectral behaviours (see Murphy et al. 2011)." -- check reference format

14. (l. 104) "In fact, although X-ray follow-up observations of far-IR selected dusty star-forming galaxies ... have clearly pinpointed the presence of a heavily obscured, accreting central supermassive black hole (see [20]), its capability of driving appreciable radio emission is still to be assessed..." -- the concordance of number would make this sentence clearer, besides correct: "In fact, although X-ray follow-up observations of far-IR selected dusty star-forming galaxies ... have clearly pinpointed the presence of heavily obscured, accreting central supermassive black holes (see [20]), their capability of driving appreciable radio emission is still to be assessed..."

15. caption of Figure 1: "DSFG" should probably be called "DSFGs"; red lines are not described in this caption; "c.l." is not defined.

16. Figure 2: the label is wrong for the points (0.01 mJy should be 0.05 mJy, and 0.05 should be 0.25mJy), to be consistent with Figure 1 (indicating 5*sigma and not rms for these surveys).

17. (l. 116): "Despite DSFGs constituting the bulk of sub-mJy radio source populations" -- a reference is missing here.

18. (l. 120) "[21] (2017, see their Fig. 8) " - the format is unconventional here, check with the editorial office.

19. (l.120) "[21] (2017, see their Fig. 8) estimated the expected contribution to the 1.4 GHz source counts of the different populations of extragalactic sources, including the overall class DSFG (tagged as SMG in their work)" -- I am confused about the population being addressed here. In [21] I understand Mancuso+17 adopt an overall SFG galaxy population -- independently of its dust content. In this paper, the DSFG population is essentially the SFG population but selected from (or bright in) FIR/millimetre wavelengths. When producing Figure 1 and Figure 2, are the authors using the SFG population from [21] to derive their detectability in the radio? How is the DSFG sub-population (relevant to this study) identified therein? I failed to find the SMG sub-population in Mancuso+2017, and I am not sure these two populations can be the same (there are dust-poor SFGs that won't be FIR/sub-mm selected but may easily appear at radio wavelengths). 

20. (l. 125) footnote to "Australia Telescope Compact Array (ATCA) at 2.1 GHz" -- in the footnote it is useful also to state the frequency range ATCA operates on.

21. (l. 135 to l. 138) "At frequencies ν ≲ 5 GHz (≳ 10 cm in wavelength) the DSFG signal is dominated by the synchrotron emission associated with star formation. The minimum, ... is expected to be located in the 30-100 GHz frequency (∼ 3 − 7 mm wavelength) range." - a reference here would be useful (e.g., Condon 92)

I feel the following paragraphs are confusing, and would deserve some re-writing:

22. (l. 156) "with a pattern that strongly correlates with the antennae configuration and the observing time allocation." -- you mean it correlates with antenna baseline length? and observing time length? 

23. (l.157) "The non-Gaussianity and the presence of noise features increases as the array is sparser or the observing time shorter, or limited to small chunks" -- I am not comfortable with this sentence. Naturally, the observation duration implies the better (or worse) sampling of the UV space, leading to higher or lower S/N and resilience to bad data. But it sounds excessive to say it in this way, and elaborating so much on something that is almost trivial. I think this entire paragraph aims to inform that radio interferometric surveys are only as good as the number of antennas (baselines) and the observation time length, and producing a wide area survey is not easy. But, as I further elaborate in the next point, we have excellent examples to the contrary.

24. (l. 171) "For all these reasons, deep radio surveys are typically limited to small areas and uncertainty due to the confusing effects significantly hampers a comprehensive description of the sub-mJy population at cm-wavelengths." -- Consider LoTSS, RACS (which the authors refer in the next paragraph), stripe82... I don't think this sentence is correct anymore. I suggest to re-word these paragraphs to reflect the current (and very exciting) deep-wide radio survey panorama.

25. (l. 189) typo in "600deg2. in five"

26. (l.192) "On the one hand, high resolution and sensitivity follow-up completed the description of the dust emission profiles" -- I don't think "completed" is a good word here. Certainly these observations helped tremendously to achieve a robust characterisation of the dust emission, but still far from "complete"...

27. (l. 195) "they clarified the gaps in the overall population description. " -- unclear which gaps this refers to...

28. (l. 200) "... to a single source..." - probably clearer to say "to a single detection". Also, since this "detection" is not really a galaxy (but a combination of several), it may be inappropriate to write "altering the determination of its dust properties and photometric redshift" but could be better to just add "confusing the determination of dust properties and photometric redshift".

29. (l. 203) "This issue should be considered together with the Herschel pointing error, which can be as bad as a few arcsec" - can this be quantified/referenced? Usually this is not a problem, as systematic offsets affecting a certain field can be (and have been) easily corrected. 

30. (l.204) "in very densely populated surveys like the IR and optical ones" - english could be polished here, as "the IR and optical" are not surveys. The author probably wants to say something like "very densely populated surveys, like those reaching very deep levels at IR and optical wavelengths"

31. (l.207) "confirmed the effects and helped de-blending" - "confirmed the effects" sounds a bit confusing, and could be dropped ("[high-res observations] helped de-blending..." )

32. (l.199 to l.217) - these paragraphs are a bit repetitive and, I feel, could be better organised and streamlined.

33. (l. 217) - the reference "Massardi et al. in preparation" should either be removed or be somewhat detailed, as it currently only states that the team is "combining deep blind radio surveys with Herschel catalogues"...

34. (l. 220) - "The availability of flux measurements in the radio and the FIR regime allows the reconstruction of the SED, hence the characterization of radio emissions, dust properties and an estimate of photometric redshift. " - The radio and FIR measurements can perhaps help in the photometric redshift determination, but I would say in a very limited way - we already know that attempts to get a FIR-radio photo-z determination are anything but accurate. Can the authors comment?

35. (l. 236) - "We performed MCMC simulations of the plausible SED fitting" - it is not clear what is meant by "plausible". Do the authors allow "synchrotron (single steep spectrum power-law), free-free (single power-law with slope equal to 0.1), and dust (grey-body spectrum) emissions" to vary freely? I guess not, which is probably how the authors decide what is "plausible" or not...?

36. (l. 241) "the SED determination for two targets " - I believe figure 3 has "four targets", not two.

37. Figure 3 is too small, mostly unreadable if printed.

38. Caption to Figure 3, "Columns 2 and 3 show the improvement in the SED definition (i.e. a thinner green region) obtained by adding only B6 or B1 ALMA respectively" - I believe column 2 adds B1 (39GHz), and column 3 adds B6 (233 GHz), which is opposite to the caption.

39. In Figure 3, why is the error bar for ALMA B1 measurement present in column 2 but not in column 4? (I believe this is not the case for the B6 measurement). Is the improvement seen due to the addition of both ALMA measurements or the reduction in the B1 measurement error?

40. Caption to Figure 3: "combination of existing ATCA and H-ATLAS data with both the proposed ALMA B1 and B6 measurements. " - it's confusing to use the term "proposed ALMA... measurements", as the authors are using actual ALMA measurements to reveal what improvement is achieved. (however, are these indeed real ALMA detections? Please clarify...)

41. Figure 4 is quite confusing, and I don't completely understand it... The B1 measurement is considered with "a signal-to-noise- ratio SNR=1 (yellow), 3 (red) and 5 (maroon)". However, I see three B1 measurements in the figure that vary not only their SNR but also their absolute flux level. I believe the authors were aiming at providing three different possibilities for the B1 measurement, *adopting the same sensitivity (noise) level* and varying the measurement flux level, leading to a varying signal for the same noise. However, this means that the authors are varying two parameters, which I am not sure is what their intent: assuming all other photometric points fixed, varying the B1 measurement level will, by itself, lead to different best-fits, even without varying the associated B1 measurement error; conversely, if one fixes the B1 measurement and varies the significance of that detection (by adopting different noise levels), the best-fits will change due to that varying measurement error alone. As it is, both the flux measurement and the significance are driving the improvement (or not) of the fit - I was expecting to see that even a low significance measurement (at the same flux level) improved the fit over the no-B1-measurement case. That could then lead to perceive that even a non-detection (upper limit only) could improve the global SED definition, which is not clear to me...

42. Also, for Figure 4, the 4 plots correspond to the same 4 sources in Figure 3, which is mentioned in the caption but should be labeled; 

43. (l. 269) "the sources seem faint in the radio domain," - I'm not sure what this sentence means and how it is mentioned in the text. The authors mention in section 3 "the sample is constituted of all the 60 sources detected at 2.1 GHz in ATCA" which reaches ~10 microJy, but that doesn't imply the sentence...

44. (l. 270) "We have demonstrated that, due to the count profiles" - should be "source count profiles"

45. (l. 272) "This approach can then be" - what approach are the authors referring to? Are they referring to "it is more efficient to attain deep sensitivities in the radio domain", implying the approach would be to focus on deep radio observations initially? Perhaps re-phrase to clarify...?

46. Figure 5 appears to be of sub-par quality, markedly different from other figures. Perhaps homogenise? 

Comments on the Quality of English Language

English is in general good, but could be improved. I include a few suggestions in the comments above.

Author Response

Response to Reviewer 1

1. (l. 29) "[DSFGs]... are the main contributors to the total star formation history (SFH) at 1 ≲ z ≲ 4 and have been detected up to z ∼ 6." - this is a strong statement that should have a reference. 
A: More references have been added.

2. (l. 33) "The presence of dust, and, thus, of high obscuration and submm/FIR luminosity, is, in fact, a signature of ongoing active star formation." - although this may be correct, the sentence ignores the possible presence of an AGN, which is found in many SMGs, and to which the authors refer immediately afterwards. I believe the sentence could be more accurately written in a more open way, such as "... is a strong indicator of ongoing active star formation, often accompanied by..."
A: corrected according to suggestion

3. (l. 37) "its emitted power through winds and/or jets will cause strong negative feedback" --- probably better as "may cause negative impact", as many cases are known of positive feedback.
A: corrected according to suggestion

4. (l. 38) "The star formation is thus quenched" --- accordingly, probably better as ""The 38 star formation may thus be quenched"
A: corrected according to suggestion

5. (l. 50) "wider beam size and the possibility of using the mosaic technique;" -- I would not say this, as (a) the "wider beam size" is, more correctly, a "wider primary beam, or field-of-view, size" and (b) the mosaic technique is not exclusive to observations in radio bands...  
A: rephrased to "it allows the construction of wide-field surveys, thanks to wider primary beams (with respect to those of submm observatories) and the possibility of combine them in mosaic mode over large areas"

6. (l. 52) "it is possible to reach high resolution (up to sub-arcsec, e.g. [6])" -- probably "(down to sub-arcsec..."?
A: corrected according to suggestion

7. (l. 54) "Radio observations ([7], [6], [8]) associated DSFGs with the faint (sub-mJy) population of objects. " -- there's a missing reference here, as I believe neither [6], nor [7] nor [8] establish any link of sub-mJy radio sources to (mm/FIR selected) DSFGs. 
A: references revised

8. (l. 58) "FIR-radio correlation (FIRRC) has been well established in the local universe ([9];[10]; [11]; [12]; [13]; [5]), " -- I don't think reference [9] has any content about the FIRRC, and neither [5]...?
A: references have been corrected

9. (l. 59) "between radio luminous AGNs and the star-forming galaxies." -- "between radio luminous AGNs and star-forming galaxies."?
A: corrected

10. (l. 60) "Thus, this relation has addressed the radio emission as an efficient tracer of obscured star formation in dusty galaxies ([14])," -- two notes here. I don't think "addressed" is the correct word here; I also don't think reference [14] is an adequate reference here. Probably you should use a reference like Condon, 1992, for using radio as a SFR (dust-free) indicator in galaxies, and then eventually provide a more recent specific reference concerning the dust-obscured population?
A: corrected according to suggestion

11. (l.69) "Despite dominant foregrounds come from our Galaxy" -- check english please.
A: we rephrased the sentence

12. (l. 72) "In this respect, masked using positions from low radio frequency surveys" -- this sentence is not clear, probably due to the english construction. Please check.
A: a portion of the sentence was missing "In this respect, once the radio loud AGNs have been masked using positions from low radio frequency surveys"

13. (l. 98) "spectral behaviours (see Murphy et al. 2011)." -- check reference format
A: corrected

14. (l. 104) "In fact, although X-ray follow-up observations of far-IR selected dusty star-forming galaxies ... have clearly pinpointed the presence of a heavily obscured, accreting central supermassive black hole (see [20]), its capability of driving appreciable radio emission is still to be assessed..." -- the concordance of number would make this sentence clearer, besides correct: "In fact, although X-ray follow-up observations of far-IR selected dusty star-forming galaxies ... have clearly pinpointed the presence of heavily obscured, accreting central supermassive black holes (see [20]), their capability of driving appreciable radio emission is still to be assessed..."
A: corrected

15. caption of Figure 1: "DSFG" should probably be called "DSFGs"; red lines are not described in this caption; "c.l." is not defined.
A: corrected

16. Figure 2: the label is wrong for the points (0.01 mJy should be 0.05 mJy, and 0.05 should be 0.25mJy), to be consistent with Figure 1 (indicating 5*sigma and not rms for these surveys).
A: the label has been corrected

17. (l. 116): "Despite DSFGs constituting the bulk of sub-mJy radio source populations" -- a reference is missing here.
A: reference added

18. (l. 120) "[21] (2017, see their Fig. 8) " - the format is unconventional here, check with the editorial office.
A: rephrased to be more conventional

19. (l.120) "[21] (2017, see their Fig. 8) estimated the expected contribution to the 1.4 GHz source counts of the different populations of extragalactic sources, including the overall class DSFG (tagged as SMG in their work)" -- I am confused about the population being addressed here. In [21] I understand Mancuso+17 adopt an overall SFG galaxy population -- independently of its dust content. In this paper, the DSFG population is essentially the SFG population but selected from (or bright in) FIR/millimetre wavelengths. When producing Figure 1 and Figure 2, are the authors using the SFG population from [21] to derive their detectability in the radio? How is the DSFG sub-population (relevant to this study) identified therein? I failed to find the SMG sub-population in Mancuso+2017, and I am not sure these two populations can be the same (there are dust-poor SFGs that won't be FIR/sub-mm selected but may easily appear at radio wavelengths). 
A: We apologize but the misunderstanding was originated by a typo here: we meant to write SFG not SMG (that in fact are never mentioned in Mancuso+17). We have now clarified that our predictions from Fig. 1 and 2 (like in Mancuso+17) refer to the overall star-forming galaxy populations. This is correct when discussing what to expect from a blind radio survey, uninformed by previous selections. Our point is that radio counts down to faint fluxes (e.g., 100 microJy) are strongly dominated by star-forming systems (radio-loud AGNs are minor contributors), and those located at z>1.5 have corresponding SFR>100 Msun/yr (see revised text for specific examples) hence will be rich in metals, dust and thus strongly obscured; in other words, practically all high-redshift star-forming galaxies selected in this way will be DSFGs. Being so rich in dust, they will have strong FIR/submm emission (as also expected basing on the radio luminosity and a standard FIR-radio correlation) and so can be easily disentagled from other low-z less star-forming and less obscured systems by cross-matching the radio survey with ancilliary FIR/submm observations. The relevant paragraphs have been rephrased to clarify our meaning and avoid possible misunderstanding.

20. (l. 125) footnote to "Australia Telescope Compact Array (ATCA) at 2.1 GHz" -- in the footnote it is useful also to state the frequency range ATCA operates on.
A: the footnote has been refined

21. (l. 135 to l. 138) "At frequencies ν ≲ 5 GHz (≳ 10 cm in wavelength) the DSFG signal is dominated by the synchrotron emission associated with star formation. The minimum, ... is expected to be located in the 30-100 GHz frequency (∼ 3 − 7 mm wavelength) range." - a reference here would be useful (e.g., Condon 92)
A: reference added

22. (l. 156) "with a pattern that strongly correlates with the antennae configuration and the observing time allocation." -- you mean it correlates with 
antenna baseline length? and observing time length? 
A: rephrased

23. (l.157) "The non-Gaussianity and the presence of noise features increases as the array is sparser or the observing time shorter, or limited to small chunks" -- I am not comfortable with this sentence. Naturally, the observation duration implies the better (or worse) sampling of the UV space, leading to higher or lower S/N and resilience to bad data. But it sounds excessive to say it in this way, and elaborating so much on something that is almost trivial. I think this entire paragraph aims to inform that radio interferometric surveys are only as good as the number of antennas (baselines) and the observation time length, and producing a wide area survey is not easy. But, as I further elaborate in the next point, we have excellent examples to the contrary.
24. (l. 171) "For all these reasons, deep radio surveys are typically limited to small areas and uncertainty due to the confusing effects significantly hampers a comprehensive description of the sub-mJy population at cm-wavelengths." -- Consider LoTSS, RACS (which the authors refer in the next paragraph), stripe82... I don't think this sentence is correct anymore. I suggest to re-word these paragraphs to reflect the current (and very exciting) deep-wide radio survey panorama.
A: we are well aware of the prodigious developments in this field in the recent years. The issue description aimed at summarizing the reasons why "radio interferometric surveys are only as good as the number of antennas (baselines) and the observation time length, and producing a wide area survey is not easy", in the spirit of reviewing observations and detection properties of DSFGs for the special issue this paper belongs to. We rephrased the paragraphs trying to provide a more optimistic perspective.

25. (l. 189) typo in "600deg2. in five"
A: corrected

26. (l.192) "On the one hand, high resolution and sensitivity follow-up completed the description of the dust emission profiles" -- I don't think "completed" is a good word here. Certainly these observations helped tremendously to achieve a robust characterisation of the dust emission, but still far from "complete"...
A:corrected as suggested

27. (l. 195) "they clarified the gaps in the overall population description. " -- unclear which gaps this refers to...
A: rephrased

28. (l. 200) "... to a single source..." - probably clearer to say "to a single detection". Also, since this "detection" is not really a galaxy (but a combination of several), it may be inappropriate to write "altering the determination of its dust properties and photometric redshift" but could be better to just add "confusing the determination of dust properties and photometric redshift".
A: corrected as suggested

29. (l. 203) "This issue should be considered together with the Herschel pointing error, which can be as bad as a few arcsec" - can this be quantified/referenced? Usually this is not a problem, as systematic offsets affecting a certain field can be (and have been) easily corrected. 
A: rephrased

30. (l.204) "in very densely populated surveys like the IR and optical ones" - english could be polished here, as "the IR and optical" are not surveys. The author probably wants to say something like "very densely populated surveys, like those reaching very deep levels at IR and optical wavelengths"
A: corrected as suggested

31. (l.207) "confirmed the effects and helped de-blending" - "confirmed the effects" sounds a bit confusing, and could be dropped ("[high-res observations] helped de-blending..." )
A: corrected as suggested

32. (l.199 to l.217) - these paragraphs are a bit repetitive and, I feel, could be better organised and streamlined.
A: paragraphs reorganized 

33. (l. 217) - the reference "Massardi et al. in preparation" should either be removed or be somewhat detailed, as it currently only states that the team is "combining deep blind radio surveys with Herschel catalogues"...
A: reference removed

34. (l. 220) - "The availability of flux measurements in the radio and the FIR regime allows the reconstruction of the SED, hence the characterization of radio emissions, dust properties and an estimate of photometric redshift. " - The radio and FIR measurements can perhaps help in the photometric redshift determination, but I would say in a very limited way - we already know that attempts to get a FIR-radio photo-z determination are anything but accurate. Can the authors comment?
A: As described in our tests, in case the dust peak is properly characterized an attempt of reconstruction of the photometric redshift could be pursued. However, this requires strong assumptions on the dust properties (temperature, components) that intrinsically imply a low accuracy in the redshift estimates. With this in mind, photometric redshifts are (trivially!) still considered the best available estimate until a spectroscopic redshift is obtained. 

35. (l. 236) - "We performed MCMC simulations of the plausible SED fitting" - it is not clear what is meant by "plausible". Do the authors allow "synchrotron (single steep spectrum power-law), free-free (single power-law with slope equal to 0.1), and dust (grey-body spectrum) emissions" to vary freely? I guess not, which is probably how the authors decide what is "plausible" or not...?
A: We perform MCMC tests, leaving the parameters free to run within ranges of possible values to identify the condition that minimizes the chi-square between the fit and the data and therefore makes the fit "plausible". The fit is obtained by summing the various emitting components for which we assumed the simplified plausible spectral behaviour described.

36. (l. 241) "the SED determination for two targets " - I believe figure 3 has "four targets", not two.
A: corrected also according to the following point

37. Figure 3 is too small, mostly unreadable if printed.
A: In order to improve the visualization we increased the panels and focussed on a single target that is representative of the whole sample.

38. Caption to Figure 3, "Columns 2 and 3 show the improvement in the SED definition (i.e. a thinner green region) obtained by adding only B6 or B1 ALMA respectively" - I believe column 2 adds B1 (39GHz), and column 3 adds B6 (233 GHz), which is opposite to the caption.
A: right, corrected

39. In Figure 3, why is the error bar for ALMA B1 measurement present in column 2 but not in column 4? (I believe this is not the case for the B6 measurement). Is the improvement seen due to the addition of both ALMA measurements or the reduction in the B1 measurement error?
A: the figure has been corrected and the caption and the text clarified accordingly. 

40. Caption to Figure 3: "combination of existing ATCA and H-ATLAS data with both the proposed ALMA B1 and B6 measurements. " - it's confusing to use the term "proposed ALMA... measurements", as the authors are using actual ALMA measurements to reveal what improvement is achieved. (however, are these indeed real ALMA detections? Please clarify...)
A: label corrected.

41. Figure 4 is quite confusing, and I don't completely understand it... The B1 measurement is considered with "a signal-to-noise- ratio SNR=1 (yellow), 3 (red) and 5 (maroon)". However, I see three B1 measurements in the figure that vary not only their SNR but also their absolute flux level. I believe the authors were aiming at providing three different possibilities for the B1 measurement, *adopting the same sensitivity (noise) level* and varying the measurement flux level, leading to a varying signal for the same noise. However, this means that the authors are varying two parameters, which I am not sure is what their intent: assuming all other photometric points fixed, varying the B1 measurement level will, by itself, lead to different best-fits, even without varying the associated B1 measurement error; conversely, if one fixes the B1 measurement and varies the significance of that detection (by adopting different noise levels), the best-fits will change due to that varying measurement error alone. As it is, both the flux measurement and the significance are driving the improvement (or not) of the fit - I was expecting to see that even a low significance measurement (at the same flux level) improved the fit over the no-B1-measurement case. That could then lead to perceive that even a non-detection (upper limit only) could improve the global SED definition, which is not clear to me...
A: The idea here was to demonstrate what is the achievement out of 10 minutes of ALMA observations (i.e. a given sensitivity) in case the absolute value would be different. However, we agree that this implies that both the absolute value and its significance are varies. Therefore we changed the image according to the reviewer suggestion, fixing the absolute value of the B1 and varying the noise between the non-detection, the 3 sigma detection and the confidence level attainable in 10 minutes on source. The same has been performed also in B6 to show how the two bands affect differently the various fitted components. 

2. Also, for Figure 4, the 4 plots correspond to the same 4 sources in Figure 3, which is mentioned in the caption but should be labeled; 
A: Now only one source is plotted both in fig 3 and 4

43. (l. 269) "the sources seem faint in the radio domain," - I'm not sure what this sentence means and how it is mentioned in the text. The authors mention in section 3 "the sample is constituted of all the 60 sources detected at 2.1 GHz in ATCA" which reaches ~10 microJy, but that doesn't imply the sentence...
A: rephrased

44. (l. 270) "We have demonstrated that, due to the count profiles" - should be "source count profiles"
A: corrected

45. (l. 272) "This approach can then be" - what approach are the authors referring to? Are they referring to "it is more efficient to attain deep sensitivities in the radio domain", implying the approach would be to focus on deep radio observations initially? Perhaps re-phrase to clarify...?
A: rephrased

46. Figure 5 appears to be of sub-par quality, markedly different from other figures. Perhaps homogenise? 
A: We changed its size to make it more similar to other images

Reviewer 2 Report

Comments and Suggestions for Authors

This paper presents a nice discussion of using a joint radio-submm-FIR approach to selecting and characterizing dusty star-forming galaxies (DSFGs) and will make a useful contribution to the literature. I have some minor comments, listed below:

Section 2.1/Figure 1

The authors have focused here on the ATCA, it would be interesting to see predictions for other current and future radio telescopes, e.g. MeerKAT/VLA/SKA-mid which should all be significantly more effective at such surveys.

Section 2.1, end of paragraph 7: modern wide-band mosaicking software such as the routines in CASA is designed to mitigate issues with varying primary beam size across the band (though it can be very computationally expensive to implement). Also bandwidth smearing is not the problem it used to be - modern correlators can correlate many narrow bandwidth channels for continuum observations. I would say for paragraph 8 that the main problem these days is to find sufficient computing resources to be able to use the most up-to-date algorithms to make deep, wide field and high dynamic range images rather than intrinsic issues with radio interferometry as a technique.

Section 3, paragraph 8. The ALMA wideband sensitivity upgrade should indeed increase the bandwidth by factors of 2-4 in bands 1 and 6, however, the gains from the digitization of the new correlator will be much smaller than a factor of 10. https://arxiv.org/pdf/2211.00195.pdf section 2.5 indicates that the improvement in sensitivity will be about 20% (40% in observing time).

Comments on the Quality of English Language

There are some small grammatical and spelling errors throughout the paper. Also many 1-sentence paragraphs that should be merged.

Author Response

Response to Reviewer 2

R: Section 2.1/Figure 1 The authors have focused here on the ATCA, it would be interesting to see predictions for other current and future radio telescopes, e.g. MeerKAT/VLA/SKA-mid which should all be significantly more effective at such surveys.

A: The analysis we performed in Sec. 3 are based on proprietary ATCA data, and for that reason we mostly focussed on survey estimates for it. Extending the predictions to other telescopes, albeit possible, is beyond the purposes of the current paper. A first order estimate of VLA condition could be obtained by dividing the observing time by a factor 25 to obtain the same survey in the same area and sensitivity conditions as estimated for ATCA, simply accounting for the difference in number of baselines. Times will be even shorter for MeerKAT (also accounting for the larger primary beams). Therefore the estimate for ATCA accounts for the current worst condition to achieve suitable submJy surveys at 2.1GHz. We stress anyway that the synergetic role of ALMA observations to complement radio and FIR data (which is the most relevant prediction here) would remain the same whatever the origin of the radio data is.   

R: Section 2.1, end of paragraph 7: modern wide-band mosaicking software such as the routines in CASA is designed to mitigate issues with varying primary beam size across the band (though it can be very computationally expensive to implement). Also bandwidth smearing is not the problem it used to be - modern correlators can correlate many narrow bandwidth channels for continuum observations. I would say for paragraph 8 that the main problem these days is to find sufficient computing resources to be able to use the most up-to-date algorithms to make deep, wide field and high dynamic range images rather than intrinsic issues with radio interferometry as a technique.
A: This is a fully agreable statement. Our comment referred mostly to "old" surveys. We hope we clarified it in paragraph 8, including the good point made by the reviewer.

R: Section 3, paragraph 8. The ALMA wideband sensitivity upgrade should indeed increase the bandwidth by factors of 2-4 in bands 1 and 6, however, the gains from the digitization of the new correlator will be much smaller than a factor of 10. https://arxiv.org/pdf/2211.00195.pdf section 2.5 indicates that the improvement in sensitivity will be about 20% (40% in observing time).
A: The Carpenter paper states (pag 12) that "For continuum observations, the upgraded Band 6 receiver will be 4.8× faster for the 2× bandwidth upgrade and 9.6× faster for the 4× bandwidth upgrade." We rephrased the sentence in section 8.

R: There are some small grammatical and spelling errors throughout the paper. Also many 1-sentence paragraphs that should be merged.
A: Some typos have been corrected and sentences merged (minor editings have not been indicated in boldface in the revised text)

Round 2

Reviewer 1 Report

Comments and Suggestions for Authors

I think the paper has improved significantly. Some clarifications/comments  remain, as indicated below:

Concerning my previous comment 10: Probably you should use a reference like Condon, 1992, for using radio as a SFR (dust-free) indicator in galaxies, and then eventually provide a more recent specific reference concerning the dust-obscured population?

This should be be the ARAA one:

doi:10.1146/annurev.aa.30.090192.003043 

Concerning my previous comment 21. a reference here would be useful (e.g., Condon 92)

Condon 1992 should be the ARAA one:

doi:10.1146/annurev.aa.30.090192.003043 

Figure 3:

Axis labels in Figure 3 have the latex symbols.

I was confused about the authors using, or not, errors for the B1 and B6 measurements. These possible detections are clearly above 100 uJy for B1 and around 1mJy for B6, so I guess they have very small errors which perhaps are neglected?

I still have concerns about Figure 4, as follows:

1) the label is incorrect in what concerns the detection from Figure 3 (maroon), as it indicates "SNR_B1=20uJy" which is incorrect (SNR doesn't even have units...). The maroon region concerns the detection, as seen from the plot, at a ~150uJy level (B1), for a sigma (noise) level of 20uJy - I think that is what is meant here. Same for the right-hand panel.

2) The caption of Figure 4 is then confusing, referring to the first two possibilities in terms of SNR and then giving just the noise for the third possibility "...and sigma = 20μJy (as in Fig 3, maroon)" - The same applies to the right hand plot for B2.

3) If I am understanding correctly, the caption could say "...three cases of B1 observations: a non-detection (upper limit at 150uJy in the example), at a 3sigma significance (150uJy +-50uJy), and a strong detection above 5 sigma significance. As a reference, a 10 minute observation with ALMA results in a noise level of ~20uJy."

4) Still, please clarify the non-detection level. Are you assuming an upper limit on your fit of 150uJy? 

(same applies to the B6 case).

Comments on the Quality of English Language

I think it is now acceptable.

Author Response

Q0. I think the paper has improved significantly. Some clarifications/comments remain, as indicated below.

A0. Thanks for the useful comments.

Q1. Concerning my previous comment 10: Probably you should use a reference like Condon, 1992, for using radio as a SFR (dust-free) indicator in galaxies, and then eventually provide a more recent specific reference concerning the dust-obscured population?
This should be be the ARAA one:
doi:10.1146/annurev.aa.30.090192.003043 

A1. Rephrased as suggested. Reference to Condon+92 ARAA fixed. 

Q2. Concerning my previous comment 21. a reference here would be useful (e.g., Condon 92)
Condon 1992 should be the ARAA one:
doi:10.1146/annurev.aa.30.090192.003043 

A2. Reference to Condon+92 ARAA fixed. 

Q3. Figure 3:
Axis labels in Figure 3 have the latex symbols.

A3. Fixed.

Q4. I was confused about the authors using, or not, errors for the B1 and B6 measurements. These possible detections are clearly above 100 uJy for B1 and around 1mJy for B6, so I guess they have very small errors which perhaps are neglected?

A4. The uncertainties are present but they are so smaller than the symbol size. 

Q5. I still have concerns about Figure 4, as follows:

1) the label is incorrect in what concerns the detection from Figure 3 (maroon), as it indicates "SNR_B1=20uJy" which is incorrect (SNR doesn't even have units...). The maroon region concerns the detection, as seen from the plot, at a ~150uJy level (B1), for a sigma (noise) level of 20uJy - I think that is what is meant here. Same for the right-hand panel.

2) The caption of Figure 4 is then confusing, referring to the first two possibilities in terms of SNR and then giving just the noise for the third possibility "...and sigma = 20μJy (as in Fig 3, maroon)" - The same applies to the right hand plot for B2.

3) If I am understanding correctly, the caption could say "...three cases of B1 observations: a non-detection (upper limit at 150uJy in the example), at a 3sigma significance (150uJy +-50uJy), and a strong detection above 5 sigma significance. As a reference, a 10 minute observation with ALMA results in a noise level of ~20uJy."

4) Still, please clarify the non-detection level. Are you assuming an upper limit on your fit of 150uJy? 
(same applies to the B6 case).

A5. We fix the label to sigma=20 muJy. We rephrase the caption as suggested. We clarify that the nondetection is actually an upper limit.